# Optic Disc Segmentation Using Attention-Based U-Net and the Improved Cross-Entropy Convolutional Neural Network

**DOI:** 10.3390/e22080844

**Published:** 2020-07-30

**Authors:** Baixin Jin, Pingping Liu, Peng Wang, Lida Shi, Jing Zhao

**Affiliations:** 1College of Computer Science and Technology, Jilin University, Changchun 130012, China; jinbx18@mails.jlu.edu.cn (B.J.); pengwang18@mails.jlu.edu.cn (P.W.); shild18@mails.jlu.edu.cn (L.S.); 2Key Laboratory of Symbolic Computation and Knowledge Engineering of Ministry of Education, Jilin University, Changchun 130012, China; 3School of Mechanical Science and Engineering, Jilin University, Changchun 130025, China; 4Department of Ophthalmology, the Second Hospital of Jilin University, Changchun 130012, China; lhbswqw@126.com

**Keywords:** information aggregation, attention mechanism, improved cross entropy, optic disc, segmentation network

## Abstract

Medical image segmentation is an important part of medical image analysis. With the rapid development of convolutional neural networks in image processing, deep learning methods have achieved great success in the field of medical image processing. Deep learning is also used in the field of auxiliary diagnosis of glaucoma, and the effective segmentation of the optic disc area plays an important assistant role in the diagnosis of doctors in the clinical diagnosis of glaucoma. Previously, many U-Net-based optic disc segmentation methods have been proposed. However, the channel dependence of different levels of features is ignored. The performance of fundus image segmentation in small areas is not satisfactory. In this paper, we propose a new aggregation channel attention network to make full use of the influence of context information on semantic segmentation. Different from the existing attention mechanism, we exploit channel dependencies and integrate information of different scales into the attention mechanism. At the same time, we improved the basic classification framework based on cross entropy, combined the dice coefficient and cross entropy, and balanced the contribution of dice coefficients and cross entropy loss to the segmentation task, which enhanced the performance of the network in small area segmentation. The network retains more image features, restores the significant features more accurately, and further improves the segmentation performance of medical images. We apply it to the fundus optic disc segmentation task. We demonstrate the segmentation performance of the model on the Messidor dataset and the RIM-ONE dataset, and evaluate the proposed architecture. Experimental results show that our network architecture improves the prediction performance of the base architectures under different datasets while maintaining the computational efficiency. The results render that the proposed technologies improve the segmentation with 0.0469 overlapping error on Messidor.

## 1. Introduction

Because the vision loss caused by glaucoma is irreversible [1], early screening for glaucoma disease is particularly important. Early detection relies on manual observation by an ophthalmologist, but it is time-consuming and laborious for each doctor to observe one by one, and the medical skills of the needed doctor are also very high. The judgment results of different doctors are also different, which is not suitable for crowd screening. Therefore, in the large-scale screening of glaucoma diseases, an automated method that saves manpower is needed. In the clinic, the cup-to-disk ratio (CDR) [2] of the fundus image is an important indicator for clinical diagnosis of glaucoma. In general, the greater the CDR, the greater the risk of glaucoma, and vice versa. Using computer technology to segment the fundus image becomes the key. The automatic segmentation method of the fundus image optic disc is mainly divided into two categories, methods based on image processing and hand-made features, and methods based on deep learning.

Image processing-based methods include threshold-based algorithms and active contour algorithms. The algorithm based on threshold makes use of the color difference of each region of the fundus image to generate binary images. Joshi et al. proposed a cup boundary detection scheme based on the appearance of pallor in Lab color space and the expected cup symmetry [3]. Cheng et al. proposed a disc segmentation method based on peripapillary atrophy elimination [4]. Noor et al. proposed a method for glaucoma detection using digital fundus images with color multi-thresholding segmentation [5]. Issac et al. used the identification parameters of glaucoma infection as features and were input into a learning algorithm for glaucoma diagnosis [6]. However, threshold-based methods are not robust enough for fundus images with low contrast or presence of pathologies [7]. The active contour algorithm divides different regions of medical images by minimizing the energy function. Joshi et al. proposed a method to integrate local image information around each point of interest in a multi-dimensional feature space [8]. However, these methods are prone to fall into the local minimum, and the performance depends largely on the model initialization. Chen et al. [9] proposed to subdivide the disc image into super pixels, and then use manual features to classify the super pixels. Wong et al. [10] proposed a method of automatic segmentation of image region by detecting vascular kinks.

Deep learning describes various computing models composed of multiple processing layers. These layers mainly learn abstract representations of different levels of data. Deep learning has powerful feature extraction capabilities. In recent years, more and more deep learning-based methods have been applied to the field of fundus image segmentation. In particular, the success of U-Net [11] has promoted the development of medical image segmentation. This network aggregates low-resolution features (providing a basis for object category recognition) and high-resolution features (providing accurate pixel positioning basis), and largely solves the problem of neglecting useful information. The research direction of fundus image segmentation methods focuses on extracting more abstract image features. CE-Net [12] uses multi-branch atrous convolution to extract features of different receptive fields. M-Net [13] uses multi-label networks and polar coordinate transformation in fundus image segmentation tasks. These methods aggregate information of different scales. After extracting features from the encoding path, high-level features fuse feature information of different scales. Zhang et al. [14] used the edge guidance module to learn the edge attention representation in the early coding layer, and then transferred it to the multi-scale decoding layer, using the weighted aggregation module fusion. Although these methods based deep learning have achieved significant results, the dependencies between channel mappings of different resolutions have been ignored.

Attention mechanism is gradually gaining popularity in medical segmentation. The attention mechanism can be viewed as using feature map information to select and locate the most significant part of the input signal [15]. Hu et al. [16] used global average pooling to aggregate feature map information, then reduced it to a single channel feature map, and finally used an activation gate to highlight salient features. Wang et al. [17] added an attention module to the residual network for image classification. Fu et al. [18] proposed a dual attention network based on spatial and channel attention mechanism. Li et al. [19] proposed a pyramid attention network that combines attention mechanisms with spatial pyramids to extract accurate features for pixel labeling. Guo et al. [20] used the residual block in the channel attention mechanism and proposed that the channel attention residual block improves the recognition ability of the network. Mou et al. [21] used a self-attention mechanism in the encoder and decoder to combine local features and global correlation. However, these attention mechanisms do not take the impact of multi-scale image features on the attention gate into account, and the channel dependence between different scales is ignored.

Inspired by the successful application of the channel attention mechanism in the field of medical image segmentation [19,20,21], we introduced an aggregation channel attention network to improve the performance of optic disc segmentation of fundus images. First, in order to alleviate the disappearance of gradients and reduce the number of parameters [22], we use DenseNet blocks to extract high-level features. Second, high-level features are more effective in classifying categories, but weaker in reconstructing the original resolution binary prediction, while low-level features are the opposite. Therefore, we propose an aggregation channel attention upsampling module, which guides the reconstruction of the original resolution by aggregating feature information of different resolutions. Third, in the task of fundus optic disc segmentation, the optic disc often occupies a small area in the image. The imbalance of the foreground and background ratio often leads to the learning process falling into the local minimum of the loss function. Dice coefficients perform well in small area image segmentation in the field of medical image segmentation [12]. To solve this problem, we combine the dice coefficients with cross entropy to balance the contribution of the two loss functions.

As illustrated above, in our paper, the main contributions to the fundus image segmentation are the following four aspects:(1)In order to avoid overfitting and save model calculation, we propose using DenseNet blocks to extract features in the encoding layer. This is particularly important in the field of medical image segmentation where data sets are generally small.(2)We propose an effective semantic segmentation decoder, called the aggregation channel attention upsampling module. We use different layers of features to guide the attention mechanism, so as to fuse the information of different scales to restore pixel categories. We use squeeze excitation blocks and generalized average pooling to integrate channel information.(3)We improved the basic classification framework based on cross entropy to optimize the network. This loss function balances the contribution of dice coefficients and cross-entropy loss to the segmentation task.(4)In order to verify the effectiveness of our method, we validated our method on the Messidor [23] and RIM-ONE [24] datasets. Compared with the existing methods, the segmentation performance of our method on these fundus image datasets has been significantly improved. This further develops the application of attention mechanism and entropy in the field of image segmentation, and promotes deep learning research in the field of optic disc segmentation of fundus images.

We would like to present the organization of our paper as follows: We give a detailed interpretation of our proposed method and the framework of the aggregation channel attention network with our method in Section 2. In Section 3, we give some details of our experiments and present their results and analysis. Lastly, we present the conclusions of our paper.

## 2. Materials and Methods

### 2.1. Aggregation Channel Attention Network Architecture for Medical Image Segmentation

As shown in Figure 1, in the encoder–decoder network structure, the encoder aims to gradually reduce the spatial size of the feature map and capture more advanced semantic features. The decoder restores the details and spatial dimensions of the object and retains more spatial information. Among the many algorithms that improve U-Net, there are improvements to the encoder and decoder, respectively. In order to obtain more significant advanced semantic features, we chose the DenseNet block that performs well in the encoder path. Similarly, in the decoding path, we propose an aggregation channel attention upsampling (ACAU) module to retain more spatial information. In order to extract contextual semantic information and generate more advanced features, in bottleneck, we use the Dense Atrous Convolution module (DAC) composed of multi-branch atous convolution and the Residual Multi-kernel pooling (RMP) composed of multi-scale pooling [12].

Figure 2 shows the proposed network structure framework. As with typical architecture for semantic segmentation, our framework, as shown in Figure 2, includes an encoder, a decoder, and a bottleneck connecting the two parts. First, the initial features of the input image are extracted through the convolution layer. The initial convolutional layer is 7 × 7 convolution with a step size of 2 and a padding of 3. In the encoder path, we used the DenseNet [22] block structure to extract image features. DenseNet block includes dense block (feature extraction) and transition block (reduced feature map size). It consists of four DenseNet blocks for different feature resolution. The bottleneck structure further extracts features at different scales through dense atrous convolution (DAC) and residual multi-kernel pooling (RMP) [12]. The decoder path is composed of four aggregation channel attention upsampling modules, which maintains the high-level features of the encoder and restores the spatial resolution of the feature map. Finally, the output feature map is subjected to deconvolution and continuous ReLU function and 3 × 3 convolution, and then processed by the sigmoid function to obtain a prediction map.

### 2.2. Dense Convolutional Network for Encoding 

In the U-Net [11] architecture, encoding is achieved through continuous convolution and pooling operations. Continuous pooling operations and convolution reduce the feature resolution to learn increasingly abstract features. This operation hinders the intensive prediction task of detailed spatial information. Maintaining high resolution requires more training resources, so there is a trade-off between saving training resources and maintaining high resolution. In order to capture more advanced features, we need to use an encoding structure that efficiently extracts advanced features and does not take up too many training resources.

In a traditional feed-forward convolutional network, the information of (l−1)th layer is transmitted to the layer l-th layer in the following form:(1)$$u_{l}=H_{l}\left(u_{l-1}\right),$$
where u is the feature map in the information flow, and H is the convolution calculation. As shown in Figure 3a, residual block [25] adds a skip-connection so that the gradient can flow directly from the later layer to the earlier layer through the identity function:(2)$$u_{l}=H_{l}\left(u_{l-1}\right)+u_{l-1}$$

As shown in Figure 3b, dense block [22] further improves the information flow between the layers, adding direct connections from any previous layer to all subsequent layers:(3)$$u_{l}=H_{l}\left(\left[u_{0},u_{1},\ldots ,u_{l-1}\right]\right).$$

This allows each layer to directly access the loss function and the gradient of the original input signal, which facilitates the training of deeper network structures. In addition, in the task of fundus optic disc segmentation, the training set size is generally small. This dense connection structure has a regularization effect, which can reduce the risk of overfitting for tasks with a small training set size.

### 2.3. Aggregation Channel Attention Upsampling Module

We now introduce the aggregation channel attention upsampling module (ACAU). Figure 4 shows the proposed ACAU module. Recently, the attention mechanism has been well applied in the field of image segmentation [19,20,21]. Squeeze and Excitation Block has also been verified to be applicable to medical image segmentation [26]. Similarly, in our proposed ACAU, in order to improve the quality of the representation generated by the network, we use Squeeze and Excitation Block [16] in each upsampling block, adaptively weight the channel, use global information, and selectively emphasize Information features, suppress useless features. Formally, $v_{l}$ is generated by shrinking $x_{l}$ through its spatial dimensions $H_{l}$ × $W_{l}$, such that the c-th channel of $v_{l}$ is calculated by:(4)$$v_{l}^{c}=F_{GAP}\left(x_{l}^{c}\right)=\frac{1}{H_{l}\times W_{l}}\overset{H_{l}}{\underset{i=1}{\sum }}\overset{W_{l}}{\underset{i=1}{\sum }}x_{l}^{c}\left(i,j\right)$$

Then, in order to take advantage of the global information in the above channel descriptor, we need to capture channel dependencies. We chose a simple gating mechanism and a sigmoid activation:(5)$$v_{o}^{c}=F_{ex}\left(v_{l},W\right)=\delta \;\left(W_{2}\sigma \left(W_{1}v_{l}\right)\right)$$
where σ refers to the ReLU function, δ refers to the sigmoid function, and $W_{1}$ and $W_{2}$ are the weights of the fully connected layer. Finally, multiply the global information with $x_{l}$ to get the weighted features:(6)$$y_{l}=v_{o}x_{l}$$

The current decoder modules lack the feature map information of different scales, and may not be conducive to pixel restoration positioning [19]. In image segmentation networks, the image features of lower layers excite informative features in a class-agnostic manner, and are better at restoring binary prediction of image resolution. Features at higher levels have more category information [16]. The main function of the decoder module is to repair category pixel positioning. We use high-level features with rich category information to weight low-level features to select accurate resolution details.

Therefore, we perform GeM pooling [27] on high-level features to provide global context information to guide low-level features. In detail, we use 1 × 1 convolution to change the number of channels of high-level features to match low-level features. The GeM pooling features descriptor is to produce an embedding of the global distribution of channel-wise feature responses [16], so that the information of the global acceptance domain of this layer is aggregated, and this information is used to guide the lower layer features. GeM pooling can be expressed as:(7)$$v_{h}^{c}=F_{GeM}\left(x_{h}^{c}\right)={\left(\frac{1}{H_{h}\times W_{h}}\overset{H_{h}}{\underset{i=1}{\sum }}\overset{W_{h}}{\underset{i=1}{\sum }}x_{h}^{c}{\left(i,j\right)}^{p_{k}}\right)}^{\frac{1}{p_{k}}}$$
where $p_{k}$ is the pooling effect parameter, and the effect of pooling can be changed by adjusting $p_{k}$ [27]. Next, the feature vector is processed by the sigmoid function and multiplied by the low-level features:(8)$$x_{out}=\delta \left(v_{h}\right)y_{l}$$

This can effectively combine feature information of different resolutions, and use high-level features to provide guidance for low-level features.

### 2.4. Improved Cross-Entropy Loss for Optic Disc Segmentation

At the end of the network, we perform a softmax operation to get the prediction map. The softmax operation is performed to ensure that the prediction result is finally mapped into the (0,1) interval, which is used to represent the probability that the pixels are the background or the disc. As the most commonly used loss function, cross-entropy loss examines each pixel independently and compares the class prediction vector with ground-truth [15]. Then, cross entropy (CE) can be defined as:(9)$$\mathrm{CE}\left(p_{i},t_{i}\right)=-\left(t_{i}\log\left(p_{i}\right)+\left(1-t_{i}\right)\log\left(1-p_{i}\right)\right)$$
where $t_{i}\in \left\{0,1\right\}$ is the groundtruth class, and $p_{i}\in$ [0, 1] is the prediction class. The dice coefficient is a measure of the overlapping area of the picture difference area, which is used to measure the difference between the prediction map and the ground-truth. It has a better effect on the measurement of small targets [12]. Dice coefficient (DC) can be defined as:(10)$$DC\left(p_{i,}t_{i}\right)=\frac{2\sum_{i=1}^{N}p_{i}t_{i}}{\sum_{i=1}^{N}p_{i}+\sum_{i=1}^{N}t_{i}}$$

To leverage dice coefficient loss to deal with imbalances and small areas, while taking the advantages of cross-entropy loss into account, we have merged two functions, which combine the advantages of the above two functions:(11)$$L=\alpha \left(-\frac{1}{N}\overset{N}{\underset{I=1}{\sum }}t_{i}\log\left(p_{i}\right)+\left(1-t_{i}\right)\log\left(1-p_{i}\right)\right)+\left(1-\alpha \right)\left(1-\frac{2\sum_{i=1}^{N}p_{i}t_{i}+S}{\sum_{i=1}^{N}p_{i}+\sum_{i=1}^{N}t_{i}+S}\right),$$
where $t_{i}\in \left\{0,1\right\}$ is the ground-truth class corresponding to each pixel, and $p_{i}\in$ [0, 1] is the pixel class prediction output by the softmax function. *N* is the number of pixels. To prevent division by zero, we use add-one smoothing [28], which adds a unity constant S to both the numerator and denominator. α controls the contribution of cross-entropy loss and dice coefficients to fusion loss.

## 3. Experiment and Results

### 3.1. Experimental Setup

#### 3.1.1. Implementation Details

We installed CUDA10.0 and CUDNN7.0 on Ubuntu 16.04 with a single 2080Ti GPU and 64 GB RAM. The experimental system is Pytorch based. As the initial network, the ImageNet-trained DenseNet was used. During the training process, we used Adam’s optimization method, using the learning rate decay set to ${\left(1-\frac{iter}{\max\_iter}\right)}^{0.9}$, and the basic learning rate was 0.0002. The input picture size is 448 × 448. The default batchsize is set to 1. The network was trained for 200 epochs on the Messidor and RIM-ONE-R1 datasets. We follow the partition in [29] to get the training and testing images in the Messidor and RIM-ONE-R1 datasets. To evaluate the segmentation performance, we used overlapping errors as the evaluation criteria:(12)$$E=1-\frac{Area\left(X\cap Y\right)}{Area\left(X\cup Y\right)}=\frac{TP}{TP+FP+FN}$$
where *X* is the predicted disc area, and *Y* is the ground-truth disc area, Area(X∩Y) represents the overlapping part of the predicted disc area and the ground-truth disc area, and Area(X∪Y) represents the union of the predicted disc area and the ground-truth disc area.

#### 3.1.2. Data Augmentation Preprocessing

Because the number of pictures in the dataset is too small, in order to avoid overfitting, we have performed data augmentation on the dataset. We perform random horizontal flip, vertical flip, and diagonal flip on each picture, so that the number of pictures in the dataset is expanded eight times as the original data. In addition, we also randomly adjust the brightness of the picture and move it left and right to further increase the effect of data augmentation.

#### 3.1.3. Dataset and Data Processing

We performed experiments on two fundus optic disc segmentation datasets: the Messidor dataset and the RIM-ONE-R1 dataset. We want to introduce these benchmark datasets as follows.

The Messidor [23] dataset was created by the Messidor project and mainly includes color images of the eye fundus. These images are obtained in routine clinical examinations, and the optic disc area is manually annotated by ophthalmologists. The image is saved in TIFF format with a resolution of 1440 × 960. We take the center of the optic disc as the picture center and crop it into a picture of size 448 × 448. According to [12,29], the dataset is randomly divided into 1000 and 200 images are used for training and testing, respectively.

The RIM-ONE [24] dataset has three sub-datasets, RIM-ONE-R1, RIM-ONE-R2, and RIM-ONE-R3, and their numbers are 169, 455, and 159. Among them, RIM-ONE-R1 has only the disc ROI area, and there are labels manually labeling the optic disc area, which are marked by five ophthalmologists. In this experiment, we use the pictures labeled by expert 1 as the training set, and the pictures labeled by the other experts as the testing set. Since the proportions of pictures in RIM-ONE-R1 are not the same, we preprocess all pictures to make them uniform in size to 448 × 448 for training and testing.

### 3.2. Ablation Study

In this section, we show the effectiveness of the proposed improvements adopted in the proposed network. We validate the dense block and ACAU module on the Messidor. In detail, we combine denseblock with U-Net and CE-Net, respectively, direct upsampling is the same as baseline, and then ACAU module is combined with U-Net and CE-Net. Downsampling is the same as baseline.

DenseNet block has a significant effect in extracting features and reducing parameters. We use it to extract more valuable features. In the experiment, we applied DenseNet block to the classic image segmentation model in order to prove its effectiveness. As shown in Table 1, the combination of DenseNet block and U-Net improves the performance from 0.055 to 0.0532, and the overlapping error of prediction decreases by 0.0018. For CE-Net, the result is improved from 0.0518 to 0.0502, and the error of prediction decreases by 0.0016. Therefore, the experiment proves that DenseNet block improves the performance of fundus image segmentation.

In our method, in order to improve the performance of image segmentation and better restore the pixel category, we propose the ACAU module. In order to show the effectiveness of retaining image information, we combined it with a classic image segmentation model and verified the segmentation effect on the Messidor dataset. As shown in Table 1, the combination of ACAU and U-Net improves the performance from 0.055 to 0.0519, and the error of prediction decreases by 0.0031. For CE-Net, the result improves from 0.0518 to 0.0496, and the error of prediction decreases by 0.0022. The experiment proves the effectiveness of ACAU for the task of fundus optic disc segmentation. The ACAU module plays a role in integrating scale features and retaining category information.

We also combined DenseNet block and ACAU into U-Net and CE-Net. For U-Net, the result was increased from 0.055 to 0.0502, the error of prediction decreases by 0.0048, and the combination of the two modules into CE-Net is ACAU-Net. The performance was improved from 0.0518 to 0.0469 and increased by 0.0049. The experiment confirmed the effectiveness of our proposed method. Bold text highlights the best results.

### 3.3. Comparison with the Baselines

In order to prove the effectiveness of our proposed model, we compare the proposed model in this paper with more advanced algorithms at this stage. Because the ORIGA dataset is not publicly available on the Internet, we use the other two datasets Messidor and RIM-ONE in CE-Net. We compared it with the method proposed by Gu et al. [12]. In addition, we compared the performance of U-Net [11] and M-Net [13] in fundus image segmentation. Similarly, we also compared with Faster RCNN method [30] and the DeepDisc method [31]. We will directly use the results obtained in their work as a reference for comparison. We compare our proposed method with the baseline in the fundus image segmentation task. Herein, we refer to the proposed aggregation channel attention network as ACAU-Net. We set the hyperparameter $p_{k}$ to 5, α to 0.5, and work number to 4, and use the Adam optimization method to optimize the model. We present the results in Table 2.

We can see from Table 2 that our aggregation channel attention network has obtained the most advanced performance on the Messidor dataset and the RIM-ONE-R1 dataset. On Messidor, we achieved the best results, an improvement of 0.0041 over CE-Net. We also achieved considerable performance on RIM-ONE-R1. The RIM-ONE-R1 dataset has five independent annotations. Compared with CE-Net, the first expert’s annotation label is improved by 0.0047, from 0.058 to 0.0533; the second group of labels is improved from 0.107 to 0.0658, and the effect is improved by 0.0412, compared with the best performing DeepDisc before; the third group of labels has increased from 0.119 to 0.0674, the error of prediction decreases by 0.0516; and the fourth group of labels is the same as the previous best effect. Although our method does not perform as well as the best results in the fifth set of labels, the overall results still show that ACAU-Net is better than CE-Net and other methods.

We also show three sample results in Figure 5 to visually compare our method with some competing methods, including U-Net and CE-Net. The image shows that our method obtained more accurate segmentation results.

### 3.4. Parameter Analysis

In this section, we analyze the hyper-parameters of aggregation channel attention network. We give more details as follows.

#### 3.4.1. Hyper-Parameter Analysis

In the process of aggregating information, it is essential to squeeze effective high-level information to obtain more distinguishing features, which plays a key role in guiding the resolution restoration of low-level features. In order to extract more distinguishing features, we use a generalized mean pooling (GeM) to integrate features. Among them, in GeM pooling, the $p_{k}$ parameter plays a role in adjusting the global pooling effect in the aggregation channel attention upsampling module. When $p_{k}$ = 1, it is average pooling, and when $p_{k}$ approaches infinity, it is maximum pooling [27]. When $p_{k}$ is other values, pooling will have different feature aggregation effects. In order to obtain the best performance in the segmentation task, we adjust $p_{k}$ as follows. Table 3 shows the performance of the segmentation experiments when the $p_{k}$ is different.

We can conclude from Table 3 that, when $p_{k}$ is less than 5, the segmentation effect gradually becomes better. Conversely, when $p_{k}$ is larger than 5, performance will decrease. When $p_{k}$ is 5, we can get the best effect of 0.0469. We will set $p_{k}$ = 5 in the experiment. When we set $p_{k}$ to 5, the feature vectors obtained by squeezing the feature map have the best guidance on the restoration resolution.

#### 3.4.2. Loss Function Contribution Parameter

Dice coefficient has a good performance in the measurement field of small area images. In order to obtain a better segmentation effect, we combined the advantages of dice coefficients and cross-entropy loss, and added the two loss functions to weigh their contributions by adjusting α. We set α between 0–1 to adjust the contribution of cross-entropy loss and dice coefficient to fusion loss. To obtain the best performance in the segmentation task, we adjust α as follows.

From Table 4, we can conclude that, when α is less than 0.5, the segmentation effect gradually becomes better. Conversely, when α is greater than 0.5, performance will decrease. When α is 0.5, we can get the best effect of 0.0469. We will set α = 0.5 in the experiment. When we set α to 0.5, the contribution of the two parts of the loss function we designed was the most reasonable, and the experiment achieved the best performance.

## 4. Discussion

In the experiment, we found that the proposed segmentation network architecture has a great advantage over the previous algorithm. It has significant characteristics, especially in the medical fundus optic disc image segmentation, that is, the medical image segmentation is not obvious. First, we verify the validity of the DenseNet block module on the Messidor dataset. The experimental results show that, in this study, the fundus optic disc segmentation task, DenseNet block, has a significant effect on extracting features and reducing parameters, and extracts more valuable features, which has improved U-Net and CE-Net. We verify the validity of the AUAU module on the Messidor dataset. The experimental results show that the ACAU module has improved the baseline method model. We use the ACAU module to collect information at different scales, fuse low-level features that are good for pixel recovery with advanced features that contain a lot of category information, and largely help restore image resolution. This provides ideas for the development of medical small area segmentation fields such as fundus optic disc segmentation and attention mechanism in the field of medical images. In the experiment, we verified the effectiveness of the loss function we used. This loss function combines the dice coefficient and cross-entropy loss to enhance the performance of the loss function in a small area. We will verify the proposed network on the Messidor dataset and the RIM-ONE dataset. The experimental results show that our network has significantly improved the efficiency of fundus optic disc segmentation compared with the previous method. This means that ACAU-Net has made a good contribution in the field of medical image segmentation, and provides a new idea to aggregate the image features of different scales to guide the attention mechanism to improve the resolution recovery accuracy. On the Messidor dataset and the RIM-ONE dataset, our experimental results are impressive, and the segmentation performance has been significantly improved.

In the clinical field, early glaucoma screening is particularly important. Early screening relies on manual observation by an ophthalmologist, which is inefficient and different doctors have different evaluation criteria. Manual observation is not suitable for mass screening. Therefore, in recent years, many automatic segmentation algorithms have emerged for segmentation of the fundus optic disc. In this article, we provide a new efficient fundus image segmentation algorithm. Compared with previous segmentation algorithms, the segmentation accuracy is improved, which facilitates the rapid diagnosis of glaucoma screening and unified evaluation criteria. Therefore, in the clinical field, our method can help doctors to make rapid diagnosis, and can unify the diagnostic standards, which liberates the energy of the doctor, and makes the contribution of deep learning in medical diagnosis worthy of attention.

## 5. Conclusions

We have proposed a new medical image segmentation model, the aggregation channel attention network, for more accurate fundus optic disc segmentation. Compared with CE-Net, we use a pre-trained DenseNet block in the encoding layer. We add the feature information of different resolutions of the decoding layer into the attention mechanism, and use the high-level feature information to guide the low-level features to preserve spatial information. Experimental results show that our method has a good effect on the task of fundus image segmentation. In a few cases, however, the experimental results are lower than previous methods. This may be due to factors such as data preprocessing and hyperparameter adjustment. In addition, due to the limitation of Denseblock, the number of channels in the network is relatively large. In future work, we will continue to design deep architectures with less computation and suitable for small datasets in the field of medical image processing, and extend the method to other medical imaging fields, such as retinal vessel segmentation, lung segmentation, and CT image segmentation.

## Figures and Tables

**Figure 1 entropy-22-00844-f001:**
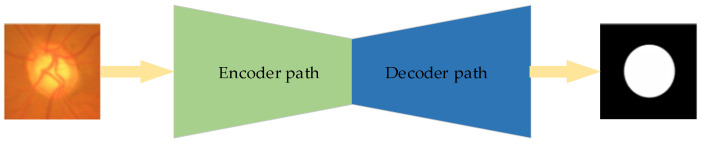
Illustration of the encoder–decoder network.

**Figure 2 entropy-22-00844-f002:**
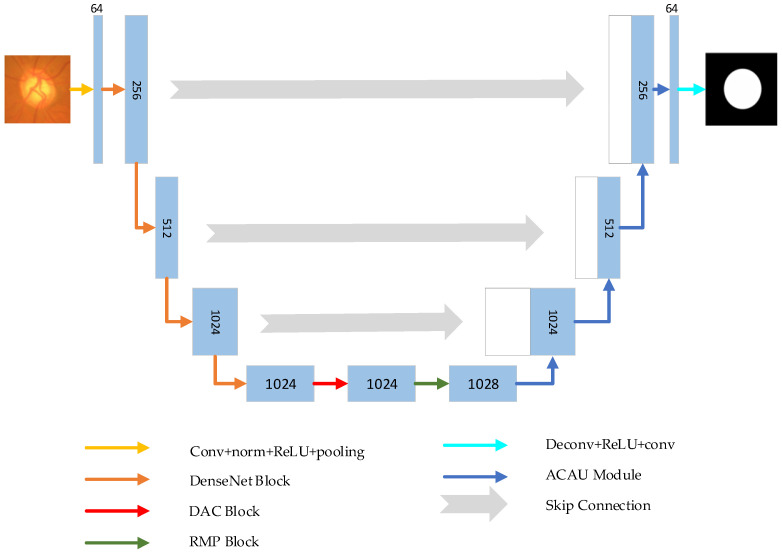
Illustration of the proposed aggregation channel attention network.

**Figure 3 entropy-22-00844-f003:**
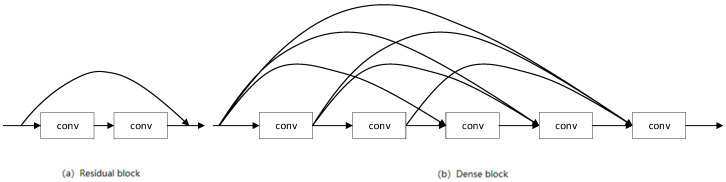
Illustration of residual block (**a**) and dense block (**b**).

**Figure 4 entropy-22-00844-f004:**
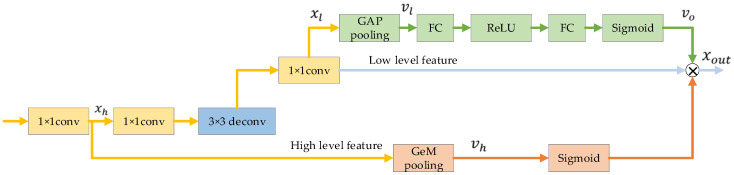
Illustration of the proposed aggregation channel attention upsampling module.

**Figure 5 entropy-22-00844-f005:**
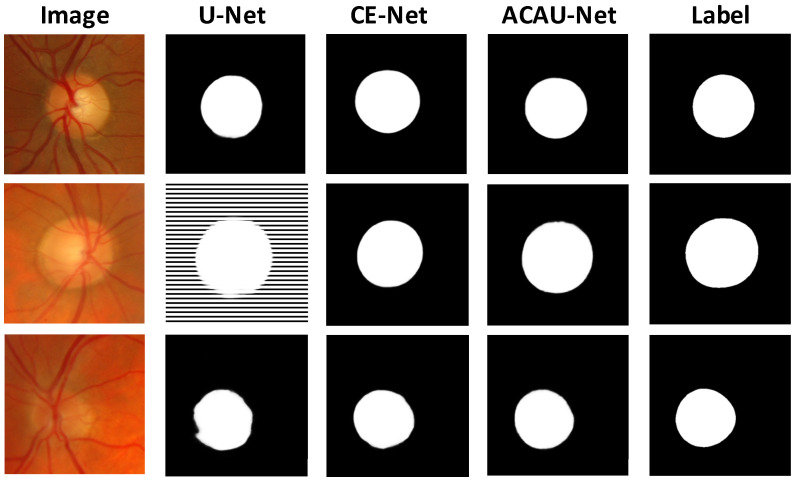
Sample results. From left to right: original fundus images, state-of-the-art results obtained by U-Net, CE-Net, ACAU-Net, and ground-truth masks.

**Table 1 entropy-22-00844-t001:** Detailed performance of Aggregation Channel Attention with different settings on Messidor. All results are achieved by us under the same experimental conditions. The best results would be highlighted in bold.

Method	E
U-Net [11]	0.055
U-Net+Denseblock	0.0532
U-Net+ACAUm	0.0519
U-Net+Denseblock+ACAUm	0.0502
CE-Net [12]	0.0518
CE-Net+Denseblock	0.0502
CE-Net+ACAUm	0.0496
ACAU-Net	**0.0469**

**Table 2 entropy-22-00844-t002:** Comparison with different methods for OD segmentation. The best results would be highlighted in bold.

Method	Messidor	R-Exp1	R-Exp2	R-Exp3	R-Exp4	R-Exp5
U-Net [11]	0.069	0.137	0.149	0.156	0.171	0.149
M-Net [13]	0.113	0.128	0.135	0.153	0.142	0.117
Faster RCNN [30]	0.079	0.101	0.152	0.161	0.149	0.104
DeepDisc [31]	0.064	0.077	0.107	0.119	0.101	0.079
CE-Net [12]	0.051	0.058	0.112	0.125	0.080	**0.059**
ACAU-Net	**0.0469**	**0.0533**	**0.0658**	**0.0674**	**0.080**	0.066

**Table 3 entropy-22-00844-t003:** The E on different *p_k_* with 3,4,5,6,7,8 on Messidor with α = 0.5. The best results would be highlighted in bold.

*p_k_*	3	4	5	6	7	8
E	0.0494	0.0495	**0.0469**	0.0487	0.0504	0.0543

**Table 4 entropy-22-00844-t004:** The E on different α with 0-1 on Messidor with *p_k_* = 5. The best results would be highlighted in bold.

α	0	0.2	0.3	0.4	0.5	0.6	0.7	1
E	0.0531	0.0517	0.0485	0.0514	**0.0469**	0.0516	0.0515	0.0522

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
