# Peer review of "Optic Disc Segmentation Using Attention-Based U-Net and the Improved Cross-Entropy Convolutional Neural Network"

_entropy, 2020, doi:10.3390/e22080844_

Round 1

Reviewer 1 Report

Dear authors.
Thank you for the chance to contribute scientifically to your article.
The following are some considerations for improving it.

Abstract.

The abstract presents a set of information that helps the reader understand what is expected of this paper.
For the abstract to be more coherent with that expected by a scientific article, the results obtained (numerical) from the image evaluation must be placed.

Introduction.

The introduction is well written. It presents the contexts to be highlighted in the paper. It presents related works.
It remains only to highlight which are the main contributions of the current paper to science.

Material and Methods

Please identify the equation variables. What do u, H. mean?

Experiment and Results

Please explain all terms in equation 12.

Conclusions.

Please highlight the strengths and weaknesses of the approach.
Expand future work.

Author Response

Point 1: For the abstract to be more coherent with that expected by a scientific article, the results obtained (numerical) from the image evaluation must be placed. 

Response 1: Thanks for your kind advice and comment for our manuscript. We added numerical results in the summary as you suggested Page 1 line 34. ‘The results render that the proposed technologies improve the segmentation with 0.0469 overlapping error on Messidor.’

Point 2: It remains only to highlight which are the main contributions of the current paper to science.

Response 2: Thanks for your helpful comments. According to the reviewer’s advice in our revised manuscript, we added clearer contribution of the article to science in Page 3 lines 123-126.

 ‘In order to verify the effectiveness of our method, we validated our method on the Messidor[23] and RIM-ONE[24] datasets. Compared with the existing methods, the segmentation performance of our method on these fundus image datasets has been significantly improved. This further develops the application of attention mechanism and entropy in the field of image segmentation, and promotes deep learning research in the field of optic disc segmentation of fundus images.’

Point 3: Please identify the equation variables. What do u, H. mean?

Response 3: We are very grateful to your kind question. In our revised manuscript, we added the meaning of the variable Page 5 line 173. ‘u is the feature map in the information flow, H is the convolution calculation.’

Point 4: Please explain all terms in equation 12.

Response 4: We are very grateful to your kind comments for our manuscript. We have explained all terms in equation 12 in Page 7 line 241. ‘Area(X∩Y) represents the overlapping part of the predicted disc area and the groundtruth disc area, Area(X∪Y) represents the union of the predicted disc area and the groundtruth disc area.’

Point 5: Please highlight the strengths and weaknesses of the approach.

Expand future work.

Response 5: Thanks for your kind advice and comment for our manuscript. We highlight the strengths and weaknesses of the approach and expand future work in Page 11 lines 398-399.  ‘In addition, due to the limitation of Denseblock, the number of channels in the network is relatively large. In future work, we will continue to design deep architectures with less computation and suitable for small datasets in the field of medical image processing, and extend the method to other medical imaging fields, such as retinal vessel segmentation, lung segmentation, and CT image segmentation.’

Reviewer 2 Report

Review for the paper :

Optic Disc Segmentation Using Attention -Based U-Net and Improved Cross-entropy Convolutional Neural Network

This article provides a new efficient fundus image segmentation algorithm. Compared with previous segmentation algorithms, the segmentation accuracy is improved, and increase the ability to rapid diagnosis of glaucoma screening.  In general the paper is well written scientifically and provides  a very detailed description of a new scheme for segmentation using deep learning technique.

I have the following comments :

  1. Page 3 line 120 : missing references for the Messidor and  RIM-ONE datasets.
  2. Page 3 line 138: it seems that the , aggregation channel attention up sampling (ACAU) is the main innovation of the manuscript. Does this term was invented by the authors ?
  3. Page 4 line 150, why the convolution block size was chosen to be 3x3 ? did you try other block sizes ?
  4. Page 4 line 153 : the title of Figure 2 should be mush shorter , the explanation of Figure 2 should be as part of the text and not as part of the title.
  5. Page 7 line 251: how many ophthalmologists were participated in the manual annotation ?
  6. Page 7 , line 258, it is indicated that "the pictures labeled by expert 1 as the training set" , what is the variance between all experts in the ground true annotation ? does this influence the results of the algorithm ?
  7. Page 8, line 280 in the description of the results the authors always write that the performance of the accuracy increase by …, while the actual number decreases for example from 0.055 to 0.0519 , I would suggest to write that the error of prediction decreases by ….this appears also in all other results.

Author Response

Point 1: Page 3 line 120: missing references for the Messidor and RIM-ONE datasets. 

Response 1: Thanks for your kind advice and comment for our manuscript. We have completed the citation of references Page 3 line 121.

Point 2: Page 3 line 138: it seems that the, aggregation channel attention up sampling (ACAU) is the main innovation of the manuscript. Does this term was invented by the authors?

Response 2: We are very grateful to your kind comments for our manuscript. In the paper, there are three innovations: the first is to apply denseblock to the coding block, the second is to propose aggregation channel attention upsampling module, which we call ACAU module, and the third is to improve cross entropy. Among them, the ACAU module is the main innovation.

Point 3: Page 4 line 150, why the convolution block size was chosen to be 3x3? Did you try other block sizes?

Response 3: Thanks for your kind advice and comment for our manuscript. At the end of our network, we need to maintain the size of the feature map, so we choose the most commonly used 3*3 convolution in convolutional neural networks to maintain the size of the feature map.

Point 4: Page 4 line 153: the title of Figure 2 should be much shorter, the explanation of Figure 2 should be as part of the text and not as part of the title.

Response 4: It is a good point, thank you very much. According to the reviewer’s advice in our revised manuscript, we simplified the title of Figure 2 and added this part of the content on page 5 lines 148-151.

Point 5: Page 7 line 251: how many ophthalmologists were participated in the manual annotation?

Response 5: We are very grateful to your kind question. Messidor is a public dataset. We checked the homepage of the dataset (http://www.adcis.net/en/third-party/messidor/ ), There is no information on how many ophthalmologists mark the optic disc area of this dataset.

Point 6: Page 7, line 258, it is indicated that "the pictures labeled by expert 1 as the training set", what is the variance between all experts in the ground true annotation? Does this influence the results of the algorithm?

Response 6: Thanks for your kind advice and comment for our manuscript. According to the RIM-ONE official document, different doctors label the ground true annotation based on their own experience. The labels are different but generally similar. Using one of the doctor's labels as the training set and the rest as the test set will not affect the results of the algorithm. In order to have an intuitive comparison with the results of CE-Net, we followed this method for experiments

Point 7: Page 8, line 280 in the description of the results the authors always write that the performance of the accuracy increase by …, while the actual number decreases for example from 0.055 to 0.0519, I would suggest to write that the error of prediction decreases by… .this appears also in all other results.

Response 7: We are very grateful to your kind comments for our manuscript. Since the overlapping error is as small as possible, it can be said to be improved. The statement suggested by the reviewer is more suitable for our expression, and we have changed everything based on your suggestion. "The overlap error of the forecast is reduced by 0.0018." All results are changed in this format.

This manuscript is a resubmission of an earlier submission. The following is a list of the peer review reports and author responses from that submission.